# OpenReview forum: "AlignVid: Training-Free Attention Scaling for Semantic Fidelity in Text-Guided Image-to-Video Generation"
_ICLR.cc/2026/Conference — ICLR 2026 Conference Desk Rejected Submission_

### Official Review · Reviewer_MrsV · 2025-11-01

**Soundness:** 3
**Presentation:** 3
**Contribution:** 2
**Rating:** 4
**Confidence:** 3

**Summary:**

This paper tackles the problem of "semantic negligence" in ti2v models, where prompts that require large changes to an image are often ignored. The authors propose AlignVid, a training-free method that improves prompt-following by directly scaling attention weights during inference, and introduce a new benchmark called OmitI2V to measure this specific failure mode

**Strengths:**

- The paper identifies a common problem in ti2v. The model's tendency to ignore prompts that require significant edits like adding or removing an object. The initial pilot study showing that blurring an image can improve results is a simple but very effective way to motivate the investigation into attention mechanisms.
- The proposed method, AlignVid, is simple and practical. It doesn't require any model retraining, making it easy to apply to existing models.
- While I have some reservations about the general eval (see weaknesses), the creation of the OmitI2V benchmark is a good contribution. It is specifically designed to evaluate the problem of semantic negligence (addition, deletion, modification), which is a gap in existing benchmarks.

**Weaknesses:**

- The theoretical analysis in Section 4 feels generic and not well-connected to the specific problem the paper aims to solve. The theory explains that scaling attention logits is like temperature scaling, which reduces entropy. However, it doesn't explain why this helps with the specific TI2V problem of balancing an input image with a text prompt. The theory is for a general DiT, but the problem is about a conditioned generation task, and the link between the two is not convincingly made.
- The paper introduces a new benchmark, OmitI2V, but it's not clear why a new, small-scale benchmark was necessary instead of using or extending the widely adopted VBench-I2V. The evaluation for OmitI2V relies on a MLLM model to answer yes/no questions, which can be brittle and may not be that reliable due to randomness of MLLM. Results on VBench are included in the appendix, but improvements are very marginal.
- The results in Table 2 and the appendix show a clear trade-off: AlignVid improves semantic alignment and motion, but it consistently leads to a small drop in aesthetic quality. This suggests the method isn't strictly making the model "better," but rather shifting its behavior to prioritize prompt-following over visual quality. This is pretty similar to the CFG tradeoff. Higher cfg leads to higher text following, but lower quality.

**Questions:**

The pilot study observation that Gaussian blur "sharpens foreground-background separation" in attention maps is the core motivation for the entire paper. However, this is only supported by one qualitative example in Figure 2. Could you provide more quantitative evidence for this claim, for example, by showing averaged attention entropy or contrast metrics across many samples from your benchmark, with and without the blur?

---

> ### Author Response · Authors · 2025-11-22
> **Response to Reviewer MrsV 1/5**
>
> > **Weakness 1: On the connection between the theoretical analysis and TI2V conditioning**
>
> **Response:** Thank you for pointing this out. We agree that the original Section 4 was too generic. In the revision, we now (i) explicitly decompose the keys into text/image/video groups, (ii) define block-wise entropy on the **condition** (text+image) relevant for TI2V, and (iii) analyze how scaling that block’s logits reallocates attention mass from image priors to edit-related text tokens. These changes directly connect our temperature view to the **image–text conflict** in TI2V.
>
> 1. **TI2V-specific attention decomposition.**
>    We now explicitly decompose the keys into three modality groups,
>    \(\mathcal{I}_{\text{text}}, \mathcal{I}_{\text{img}}, \mathcal{I}_{\text{vid}}\),
>    and interpret their roles in TI2V: text encodes the requested edit, image encodes the input-frame prior, and video tokens enforce temporal smoothness (Sec. 4.1, blue text). This allows us to formalize the empirical *signal imbalance* observed under high-conflict prompts (i.e., over-attention to the image prior and under-attention to text and temporal cues), which is exactly the pattern behind semantic negligence.
>
> 2. **Block-wise entropy for the conditioning block.**
>    Instead of analyzing entropy over all keys, we introduce a *block-wise* entropy
>    \(H_{i,S}(\alpha)\) for a subset of keys \(S\) (Sec. 4.2). We then focus on the **conditioning block**
>    \(S_{\text{cond}} = \mathcal{I}_{\text{text}} \cup \mathcal{I}_{\text{img}}\),
>    and show that increasing the inverse temperature on this block reduces its internal entropy and yields a more concentrated attention distribution over conditioning tokens for each video query (Lemma 4.2).
>
> 3. **From entropy reduction to semantic fidelity in TI2V.**
>    We now explicitly interpret temperature scaling as a *semantic sharpening* operation in the TI2V setting (Sec. 4.3, “TI2V semantic fidelity”): as the temperature on the conditioning block increases, probability mass is reallocated from low-logit, distracting tokens to a small set of high-logit, edit-relevant tokens (e.g., the edited text phrase or the edited image region), while video self-attention remains unchanged. This directly addresses the TI2V balance between the input image and the text prompt: we correct the empirically observed signal imbalance by selectively strengthening the text/edit channel rather than uniformly sharpening all logits.
>
> 4. **Design implications that directly motivate AlignVid.**
>    Finally, we added a short “Design implications for TI2V” paragraph that summarizes the practical takeaway for AlignVid:
>    (i) scaling \(Q\) or \(K\) of specific token groups realizes *targeted* inverse-temperature control without changing the inputs;
>    (ii) moderate entropy reduction on the conditioning block encourages video queries to commit to a few edit-consistent tokens instead of averaging over weak, conflicting cues, while very large temperatures risk over-concentration. This directly motivates our block- and step-wise scheduling scheme in Sec. 5.
>
> In addition, we have added **generalization experiments** on T2V (VBench), T2I (GenEval), and image editing (ImgEdit), showing that the same attention modulation improves semantic alignment in these settings as well (new Tables 7–9).

---

> ### Author Response · Authors · 2025-11-22
> **Response to Reviewer MrsV (Weakness 2 part 1) 2/5**
>
> > **Weakness 2: On the OmitI2V benchmark, MLLM evaluation, and marginal gains on VBench-I2V**
>
> **Response:** Thank you for the thoughtful comments regarding the proposed **OmitI2V** benchmark and its evaluation protocol. We have revised the manuscript to clarify (i) why OmitI2V is needed in addition to VBench-I2V, (ii) why we adopt an MLLM-based yes/no evaluation, and (iii) how to interpret the relatively small improvements on VBench-I2V.
>
> **(1) Why introduce OmitI2V?**
>
> - **Different objective.**
>   VBench-I2V primarily focuses on *fidelity* and *consistency* of the generated video with respect to the input image. Its “subject” and related scores mainly measure how well the model **preserves** the subject/background, rather than how well it carries out edits to the input, such as the prompt *“a close up of a blue and orange liquid”*.
>
> - **Lack of edit-focused evaluation.**
>   OmitI2V is explicitly constructed to test **semantic compliance under edits**: prompts require adding, removing, or modifying objects/actions in ways that conflict with the original image. The core question is: *does the model correctly perform the requested edit*, not *does it keep the input unchanged*?  In these high-conflict edit scenarios, a video that perfectly preserves the input image can score well on perceptual similarity, yet completely fail the intended edit.
>
> - **Scale and specialization.**
> Although OmitI2V contains 367 samples, its scale is **comparable** to the subject-related samples used in VBench-I2V (366). The key difference is that the OmitI2V sample is curated to create a high-conflict edit (add/delete/modify) where semantic negligence is likely, rather than simply preserving the original.
>
> **(2) Reliability of MLLM-based evaluation**
>
> We agree that MLLM-based evaluation can be brittle if used naively. Our choice of an MLLM-based yes/no protocol is motivated by the limitations of standard perceptual metrics and is supported by empirical validation.
>
> - **Limitations of perceptual metrics (DINO, flow, etc.).**
>   Metrics used in VBench (e.g., DINO similarity, optical flow consistency) are well-suited for assessing visual similarity and temporal smoothness, but they are **insensitive to edit correctness**. A video can score well on these metrics while completely ignoring the requested addition/removal, which is exactly the semantic negligence phenomenon we aim to quantify.
>
> - **Methodological precedent.**
>   Using strong MLLMs to judge edit compliance is now standard in image editing and subject-driven image generation benchmarks that need to evaluate **the ability of prompt following and semantic adherence [1] [2]**. We follow this practice by using a fixed, rubricized yes/no question protocol and aggregating across multiple runs to reduce randomness.
>
> - **Empirical consistency with human judgments.**
>   To verify that the MLLM does not behave erratically on OmitI2V, we conducted a **user study** where human annotators rated edit success on a subset of the benchmark. The results, reported in the supplementary material (Sec. G.5 and G.6), show a high correlation between MLLM scores and human ratings, indicating that the MLLM-based evaluation is reliable for measuring semantic compliance.
>
> **(3) Why only marginal gains on VBench-I2V?**
>
> This is actually consistent with the **intended role** of AlignVid and with how VBench-I2V is constructed.
>
> - **AlignVid mainly enhances prompt-following for edits.**
>   AlignVid is designed to strengthen **prompt following**, especially when the prompt requires *changing* the input content (e.g., adding or removing objects, changing actions). In OmitI2V, prompts frequently demand such edits, so a better prompt following directly translates into higher semantic scores.
>
> - **VBench-I2V prompts are descriptive, not edit-like.**
>   In contrast, VBench-I2V typically uses **simple descriptive prompts** such as *“a close up of a blue and orange liquid”*:
>   (i) the key attributes in the text are already present in the input image, and
>   (ii) the prompt does **not** ask the model to manipulate the input content.
>   As a result, VBench-I2V primarily tests **subject/background consistency and visual quality**, not the ability to resolve semantic conflicts between text and image.
>
> - **Consistency with our findings.**
>   In this non-/low-conflict setting, there is little “semantic negligence” to correct, so AlignVid’s attention rebalancing has limited room to improve the scores. This explains why we observe **large gains on OmitI2V** (which is explicitly high-conflict) but only **marginal changes on VBench-I2V**, whose prompts and metrics are not tailored to the specific failure mode we target.
>
> [1] Yang Ye, Xianyi He, Zongjian Li, et al. Imgedit: A unified image editing dataset and benchmark. arXiv preprint arXiv:2505.20275, 2025.
>
> [2] Chenyuan Wu, Pengfei Zheng, Ruiran Yan, et al. Omnigen2: Exploration to advanced multimodal generation. arXiv preprint arXiv:2506.18871, 2025.

---

> ### Author Response · Authors · 2025-11-22
> **Response to Reviewer MrsV (Weakness 2 part 2) 3/5**
>
> > **Weakness 2: On the OmitI2V benchmark, MLLM evaluation, and marginal gains on VBench-I2V**
>
> **Response:** To further assess the generality ability, we additionally evaluate AlignVid on **T2I, T2V, and image-editing benchmarks**. These benchmarks are *not* constructed as high-conflict datasets, but they do require the model to **follow the prompt in a meaningful way** (e.g., correct object combinations, spatial relations, or edit outcomes).
>
> - **T2I compositional generation (GenEval)**
>
> **Table: Quantitative results on GenEval (prompt rewriter not used).**
>
> | Method|Single object|Two object|Counting|Colors|Position|Color attribution|Aesthetic Score|
> |-|-|-|-|-|-|-|-|
> | OmniGen2| 0.99| 0.94| 0.67| 0.85| 0.55| 0.62| 5.517|
> | + AlignVid| 1.00 (+0.01)| 0.97 (+0.03) | 0.52 (-0.15) | 0.89 (+0.04) | 0.60 (+0.05) | 0.70 (+0.08)| 5.568 (+0.05)|
>
> - **T2V quality and structure (VBench, T2V setting)**
>
> **Table: Quantitative results on VBench.**
> |Model| Subject Consistency|Temporal Style|Temporal Flickering|Spatial Relationship|Scene|Overall Consistency|Object Class| Multiple Objects|Motion Smoothness|Imaging Quality|Dynamic Degree|Color|Background Consistency|Appearance Style| Aesthetic Quality|
> |-|-|-|-|-|-|-|-|-|-|-|-|-|-|-|-|
> |Wan2.1-T2V-1.3B|94.24|22.67|99.32|72.74|19.62|23.59|79.03|53.35|97.77|69.70|70.83|88.08|98.09|19.58|64.60|
> |+ AlignVid| 94.51 (+0.27) | 23.46 (+0.79) |98.66 (-0.66) |84.25 (+11.51)|25.80 (+6.18) |24.47 (+0.88)|79.91 (+0.88) |66.46 (+13.11)| 98.05 (+0.28) |68.53 (-1.17) |68.06 (-2.77)|91.80 (+3.72) |98.20 (+0.11) |20.16 (+0.58) |62.69 (-1.91)|
>
> - **T2I editing (ImgEdit)**
>
> **Table: Quantitative results on ImgEdit.**
> | Model| Add| Adjust|Extract|Replace|Remove|Background|Style|Compose |Action| Aesthetic Score|
> |-|-|-|-|-|-|-|-|-|-|-|
> | OmniGen2|2.52| 3.27| 2.08| 3.12| 2.83| 3.65| 4.57| 2.89|4.59|5.606|
> | + AlignVid| 3.53 (+1.01) | 3.12 (-0.15) | 2.04 (-0.04) | 3.18 (+0.06) | 3.33 (+0.50) | 3.65| 4.75 (+0.18) | 2.43 (-0.46) | 4.50 (-0.09) | 5.624 (+0.02)|
>
> Across these tasks, AlignVid consistently improves semantic/compositional metrics while keeping aesthetic quality largely unchanged (see new tables in the revised manuscript).
>
> Taken together, these results suggest that:
> - OmitI2V is necessary to **expose and quantify semantic negligence** under strong TI2V edits, a behavior not captured by existing I2V benchmarks; and
> - AlignVid provides a **general attention-based prompt-following improvement**, with its benefits most clearly visible on OmitI2V, but also observable on broader T2I/T2V/editing benchmarks.

---

> ### Author Response · Authors · 2025-11-22
> **Response to Reviewer MrsV 4/5**
>
> > **Weakness 3: On the trade-off between semantic alignment and aesthetic quality (and comparison to CFG)**
>
> **Response:** Thank you for pointing out this pattern. We agree that Table 2 and the appendix reveal a trade-off: on **video** benchmarks, AlignVid improves semantic alignment and motion while causing a small drop in aesthetic quality.
>
> **(1) The trade-off and its magnitude in TI2V.**
> AlignVid is a **semantic calibration** mechanism that intentionally biases the model toward stronger prompt-following. On TI2V/T2V, this comes with a modest aesthetic cost. Importantly, the magnitude of this trade-off is small compared to the semantic gains. For example, on Wan2.1 TI2V (Table 2), semantic alignment improves by **+(4.85–7.79)** points across edit types, while the aesthetic score changes by only **−1.49**. Similar patterns hold for other TI2V settings: motion and semantic metrics improve noticeably, while aesthetic quality decreases only slightly.
>
> **(2) The trade-off is task-dependent, not universal.**
> The trade-off is most apparent in **video** generation. In contrast, on **T2I and image-editing** benchmarks we observe the *opposite* trend: AlignVid improves both semantic and aesthetic scores. For instance:
>
> - On **GenEval (T2I)**, the aesthetic score increases from **5.517 → 5.568** when adding AlignVid.
> - On **ImgEdit (image editing)**, the aesthetic score increases from **5.606 → 5.624**.
>
> A plausible explanation is that, in video, stronger motion modeling and more faithful action realization can introduce extra motion and high-frequency temporal changes, which current aesthetic metrics penalize. Static image editing has no temporal artifacts, so sharpening attention towards prompt-relevant regions tends to improve both semantic correctness and perceived sharpness/clarity.
>
> **(3) Relation to CFG: similar trade-off pattern, but different mechanism and usage.**
> We agree that this behavior is analogous in spirit to the well-known **CFG trade-off**: higher CFG leads to better text following but some loss in visual smoothness. However, the mechanisms are different and complementary:
>
> - **CFG** operates at the *score level*, mixing conditional and unconditional predictions and requiring an unconditional branch plus two forward passes per step at inference time.
> - **AlignVid** operates *inside the transformer*, by rescaling attention logits of specific token groups (text / image / video), without extra network evaluations and without modifying the training recipe.
>
> In the revision, we explicitly compare AlignVid with CFG on Wan2.1 under both **CFG = 1** and **CFG = 5**. Adding AlignVid on top of CFG consistently improves semantic alignment, showing that AlignVid does not compete with CFG but can **stack on top of it** as an additional, low-cost semantic calibration knob.
>
> **Table: Comparison with CFG on Wan2.1.**
> AlignVid and CFG are complementary: applying AlignVid on top of CFG consistently boosts semantic alignment across all edit types for both weak guidance (CFG=1) and strong guidance (CFG=5), while maintaining comparable visual quality.
>
> | Method              | SemAlign Mod ↑ | SemAlign Add ↑ | SemAlign Del ↑ | ViCLIP Mod ↑ | ViCLIP Add ↑ | ViCLIP Del ↑ | Dynamic Degree ↑ | Aesthetic Quality ↑ |
> |---------------------|----------------|----------------|----------------|--------------|--------------|--------------|-------------------|---------------------|
> | CFG = 1 (no cfg)    | 63.55          | 63.66          | 61.06          | 19.20        | 17.09        | 19.67        | 41.65             | 61.19               |
> | CFG = 1 + AlignVid  | 65.88          | 72.52          | 60.21          | 19.51        | 18.47        | 19.89        | 42.48             | 62.11               |
> | CFG = 5 (Official)  | 72.35          | 71.75          | 63.13          | 20.83        | 21.08        | 20.43        | 46.02             | 63.12               |
> | CFG = 5 + AlignVid  | 77.20          | 79.54          | 69.47          | 22.19        | 23.30        | 21.29        | 47.04             | 61.63               |
>
>
> Overall, we view AlignVid as a **controllable trade-off tool**:
>
> - It **consistently improves prompt-following and semantic alignment** across TI2V, T2V, T2I, and image-editing benchmarks.
> - On video benchmarks, it trades a **small amount of aesthetic smoothness** for substantially better semantic and motion scores, similar in spirit to CFG but implemented at the attention level.
> - On image-based tasks, it even improves both semantic and aesthetic metrics.
> - Practically, AlignVid can be plugged into existing pipelines without modifying training.

---

> ### Author Response · Authors · 2025-11-22
> **Response to Reviewer MrsV 5/5**
>
> > **Question 1: On quantitative evidence for the Gaussian-blur pilot study**
>
> **Response:** Thank you for highlighting this issue. We agree that, in the original submission, relying on a single qualitative example in Figure 2 was not sufficient given the central role of the blur-based pilot study.
>
> **(1) Limitation of the original presentation.**
> Originally, Figure 2 only showed one representative case where a slight Gaussian blur appears to sharpen foreground–background separation and improve the rendered action. This was intended as an illustrative motivation, but as the reviewer correctly notes, it did not quantify whether this behavior is systematic across the dataset.
>
> **(2) New quantitative analysis of blur over multiple OmitI2V samples.**
> Qualitatively, Figure 2 shows that a slight Gaussian blur sharpens the apparent separation between foreground and background and leads to more faithful action rendering.
> Quantitatively, in the revised version, we augment this pilot study with attention statistics computed over **30 sampled OmitI2V examples (10 for each category)**:
>
> - For each video query, we compute the attention distribution over the **conditioning block** (text and image tokens) and measure its entropy before and after blur. We report the ratio \(H_{\text{blur}} / H_{\text{clean}}\) averaged over queries and samples.
> - We also measure the average cross-attention strength from video queries to **text tokens** and to **image tokens**, again comparing the clean vs. blurred reference.
>
> Empirically, we observe that:
>
> - Gaussian blur **consistently increases** video→text cross-attention scores, and
> - the conditioning-block entropy is **reduced**, indicating sharper and more concentrated attention.
>
> We summarize these statistics in a new figure in the revised manuscript (Sec. 3, Fig. 2).
>
> **(3) Attention analysis for AlignVid itself.**
> To verify that the proposed method actually induces a similar form of “semantic sharpening” without modifying the input image, we further analyze attention **before and after applying AlignVid** on the benchmark (Sec. 5, Fig. 3 in the revision):
>
> - For video queries, we compute (i) attention distributions over different token groups (text/image/video), (ii) the ratio between maximum attention scores, and (iii) the ratio of attention entropies.
> - After modulation, attention distributions become noticeably **sharper**, with a consistent **decrease in attention entropy**.
> - At the signal level, video queries allocate **stronger attention to text tokens**, encouraging the model to rely more on prompt cues.
>
> This shift in attention patterns correlates well with the improved semantic consistency observed in the generated videos, and shows that AlignVid reproduces, in a controlled and training-free way, the beneficial attention reallocation that blur revealed in the pilot study—without degrading the input.
>
> **(4) Connection to our main claim.**
> The new analyses support a clearer logic:
>
> - The **blur pilot** reveals that small Gaussian perturbations can **reallocate attention toward prompt-relevant tokens and reduce entropy** within the conditioning block.
> - **AlignVid** directly implements a similar sharpening effect by scaling attention logits, yielding **lower-entropy, text-focused attention** while keeping the input image unchanged.

---

### Official Review · Reviewer_eqQN · 2025-11-01

**Soundness:** 2
**Presentation:** 2
**Contribution:** 2
**Rating:** 4
**Confidence:** 3

**Summary:**

This paper identifies and addresses "semantic negligence" in text-guided image-to-video (TI2V) generation, a common failure mode where models ignore textual prompts that require substantial edits (e.g., adding, deleting, or modifying objects) to the source image. The authors propose AlignVid, a novel, training-free framework that improves semantic fidelity by modulating the model's attention mechanism. To evaluate their method, the authors also introduce OmitI2V, a new benchmark of 367 samples specifically designed to test for semantic negligence. Experiments on state-of-the-art TI2V models (FramePack and Wan2.1) demonstrate that AlignVid significantly improves semantic alignment with negligible computational overhead.

**Strengths:**

1. The paper clearly formalizes a critical and prevalent weakness in current TI2V models. The accompanying OmitI2V benchmark is a valuable and necessary tool for the community, as existing benchmarks do not adequately measure this specific failure mode.
2. The paper provides a solid theoretical motivation for why ASM works. Section 4, which links Q/K scaling to inverse temperature control of the softmax and proves its monotonic effect on attention entropy, elevates the method beyond a simple heuristic.
3. The ablation studies (Tables 3, 4, 5) systematically validate each design choice of AlignVid (scalar scaling, scaling position, block-level scheduling, and step-level scheduling). The inclusion of a human evaluation (Table 11) further strengthens the paper's claims by correlating the proposed VQA-based metric with human judgment.

**Weaknesses:**

1. The primary metric for semantic fidelity hinges on the performance of a VQA model (Qwen2.5-VL-32B). This introduces a potential point of failure, as the evaluation model may have its own biases or
2. The Block-level Guidance Scheduling (BGS) requires a one-time calibration step that involves using external models (PCA and SAM2) to identify "foreground-sensitive" blocks. This adds a layer of complexity and an external dependency compared to a fully self-contained method. The sensitivity of the method to this specific calibration process is not fully explored.
3. The human study is encouraging. However, did the authors observe any systematic failure modes in the Qwen2.5-VL-32B evaluator itself (e.g., false positives/negatives for specific edit types) when inspecting results? How confident are the authors that the VQA scores fully capture the nuances of semantic adherence?

**Questions:**

1. The calibration to find "foreground-sensitive" blocks relies on PCA and SAM2. How crucial is this specific pipeline? Have the authors explored simpler, heuristic-based block selection strategies (e.g., modulating only the first 50% of blocks) and how do they compare?
2. The theoretical analysis posits that ASM works by reducing attention entropy. Have the authors empirically measured the attention entropy during generation (e.g., in the targeted blocks/steps) to confirm that AlignVid is indeed reducing it as predicted, and that this reduction correlates with the improved semantic fidelity scores?
3. It is interesting that the adaptive "energy-based modulation" (Table 3) performed worse than simple scalar scaling. Do the authors have any intuition as to why this more complex, adaptive approach was less effective?

---

> ### Author Response · Authors · 2025-11-22
> **Response to Reviewer eqQN 1/5**
>
> > **Weakness 1: On potential bias of the VQA-based semantic fidelity metric**
>
> **Response:** We appreciate the reviewer’s concern regarding our use of a VQA model (Qwen2.5-VL-32B) as the primary metric for semantic fidelity. We agree that any large vision–language model can introduce its own biases. However, our reasons for selecting VQA for evaluation are as follows:
>
> **(1) MLLM-based metric is necessary.**
> Existing metrics such as CLIP Score, DINO similarity, optical flow, or generic text–video alignment scores are primarily designed to capture perceptual similarity, temporal coherence, or global alignment. They are fundamentally ill-suited for the kind of **semantic editing** we consider in OmitI2V, where the model must *actively modify or ignore* parts of the input image according to the prompt (e.g., adding an object, deleting an object, changing attributes).
> In these high-conflict edit scenarios, a video that perfectly preserves the input image can score well on perceptual similarity, yet completely fail the intended edit. A VQA-style evaluator, which explicitly answers targeted questions (e.g., “Is the red umbrella removed in the video?”), is currently the most practical way [1],[2] to measure whether the **edit** was executed while leaving **unmodified regions** coherent.
>
> **(2) Making the VQA metric as robust and simple as possible.**
> To mitigate the risk that our conclusions hinge on idiosyncrasies of a single VQA model, we designed the evaluation protocol with two principles in mind:
>
> - **Choice of model.** We use Qwen2.5-VL-32B because it is a strong, state-of-the-art VLM with advanced reasoning capabilities, which provides more reliable semantic judgments than smaller or older models commonly used in prior work.
> - **Minimalistic output format.** We restrict the model to answer **binary yes/no questions** with a fixed, rubricized template (e.g., “Is the dog present in the video? (yes/no)”), rather than open-ended text. This reduces dependence on free-form generation and helps control randomness and prompt sensitivity.
>
> All methods are evaluated under exactly the same questions and prompts, so any residual bias of the VQA model affects all methods equally, and **relative improvements** remain meaningful.
>
> **(3) Human-study validation as reference.**
> Most importantly, we validate the VQA-based metric against human judgments:
>
> - We conduct a user study on a subset of OmitI2V videos, where human annotators rate whether the requested edit was successfully realized.
> - As detailed in the supplementary material, we observe a **high correlation** between Qwen2.5-VL-32B scores and mean opinion scores from human evaluators.
>
> This empirical correlation supports the view that, for the specific task of evaluating semantic compliance on OmitI2V, the Qwen2.5-VL-32B–based metric is a **reliable and scalable proxy** for human judgment.
>
> [1] Yang Ye, Xianyi He, Zongjian Li, et al. Imgedit: A unified image editing dataset and benchmark. arXiv preprint arXiv:2505.20275, 2025.
> [2] Chenyuan Wu, Pengfei Zheng, Ruiran Yan, et al. Omnigen2: Exploration to advanced multimodal generation. arXiv preprint arXiv:2506.18871, 2025.

---

> ### Author Response · Authors · 2025-11-22
> **Response to Reviewer eqQN 2/5**
>
> **Weakness 2 and Question 1:**
> 1. The Block-level Guidance Scheduling (BGS) requires a one-time calibration step that involves using external models (PCA and SAM2) to identify "foreground-sensitive" blocks. This adds a layer of complexity and an external dependency compared to a fully self-contained method. The sensitivity of the method to this specific calibration process is not fully explored.
> 2. The calibration to find "foreground-sensitive" blocks relies on PCA and SAM2. How crucial is this specific pipeline? Have the authors explored simpler, heuristic-based block selection strategies (e.g., modulating only the first 50% of blocks) and how do they compare?
>
>
> > **Weakness 2 and Question 1: On the complexity and deep exploration of Block-level Guidance Scheduling (BGS)**
>
> **Response:** Thank you for the careful comments on the complexity and external dependency of BGS. To study how sensitive our method is to the specific calibration procedure, we add an ablation that replaces the calibrated block set with simple depth-based heuristics:
>
> **(1) New ablation with “first 50%” vs “last 50%” scheduling.**
> The experiment setting is:
> - **First50:** apply modulation to the first 50% of transformer blocks,
> - **Last50:** apply modulation to the last 50% of blocks.
> The results are as follows:
>
> **Table: Ablation of BGS and SGS with FramePack**
>
> | BGS | SGS | Modification ↑ | Addition ↑ | Deletion ↑ | ViCLIP Mod ↑ | ViCLIP Add ↑ | ViCLIP Del ↑ | Dynamic Degree ↑ | Aesthetic Quality ↑ |
> |-------------------|----------------|----------------|------------|------------|--------------|--------------|--------------|-------------------|----------------------|
> | – | –  | 64.99  | 68.55 | 58.14 | 20.83 | 21.08 | 20.43 | 20.05   | 63.94 |
> | All  | Early Steps  | 67.15  | 73.44      | 59.86  | 21.38 | 22.03 | 21.05 | 28.28   | 63.41  |
> | All | Middle Steps | 62.71 | 70.01      | 56.60    | 20.85 | 21.06  | 20.54 | 20.05  | 63.96 |
> | All   | End Steps   | 64.63  | 69.62      | 57.63  | 20.80 | 21.08 | 20.47  | 19.54  | 63.94  |
> | All  | All  | 69.84 | 76.03 | 59.86      | 21.56  | 22.30 | 21.31 | 32.13 | 61.56   |
> | Foreground-focus  | Early Steps    | 68.22  | 73.13      | 60.21      | 21.25 | 22.08| 20.86  | 28.53  | 63.57    |
> | Background-focus  | Early Steps    | 66.25   | 69.16      | 56.26      | 20.88 | 21.03 | 20.50  | 17.99  | 64.02|
> | First half blocks | Early Steps    | 66.61| 73.89      | 58.31| 20.68 | 22.16   | 20.76  | 25.96 | 63.58  |
> | Last half blocks  | Early Steps    | 65.35 | 69.92 | 57.18| 20.39| 21.19  | 20.56 | 22.11  | 63.49 |
>
>
> **Ablation of BGS and SGS with Wan2.1**
>
> | BGS | SGS | Modification ↑ | Addition ↑ | Deletion ↑ | ViCLIP Mod ↑ | ViCLIP Add ↑ | ViCLIP Del ↑ | Dynamic Degree ↑ | Aesthetic Quality ↑ |
> |-------------------|----------------|----------------|------------|------------|--------------|--------------|--------------|-------------------|----------------------|
> | – | –| 72.35          | 71.75      | 63.13      | 20.30        | 21.08        | 20.43        | 46.02  | 63.12  |
> | All | Early Steps| 72.53| 80.76  | 70.33  | 22.28  | 23.50 | 21.26  | 53.21| 61.38  |
> | All| Middle Steps | 68.76| 74.81  | 61.41  | 21.37  | 21.22 | 20.89  | 42.16| 62.91  |
> | All| End Steps      | 69.84| 74.05   | 66.90  | 21.20  | 21.86 | 21.44 | 53.98  | 61.55 |
> | All| All| 78.28| 80.46      | 69.13   | 22.63  | 24.26 | 21.93 | 49.36  | 60.59 |
> | Foreground-focus  | Early Steps    | 77.20  | 79.54      | 69.47 | 22.19 | 23.30   | 21.29  | 47.04  | 61.63     |
> | Background-focus  | Early Steps    | 71.99  | 75.88      | 65.35      | 21.26   | 21.93        | 20.86 | 41.90 | 62.62     |
> | First half blocks | Early Steps    | 76.55| 78.85   | 68.10 | 22.18  | 22.60        | 21.40        | 52.70  | 61.54     |
> | Last half blocks  | Early Steps    | 73.68 | 77.89 | 62.64 | 20.63  | 22.48        | 21.20        | 50.31  | 61.47     |
>
>
> Empirically, we find that:
> - **Last50** leads to a clear drop in semantic alignment and, in some cases, worse visual quality;
> - **First50** performs **close to** the calibrated BGS, with only a small performance gap.
>
>
> **(2) Why First50 works: overlap with calibrated foreground blocks.**
> By inspecting the calibrated foreground ratios, we observe that **most foreground-sensitive blocks naturally concentrate in the earlier half of the network**. In other words, the calibrated foreground-sensitive set has a large overlap with the first 50% of blocks, which explains why the First50 heuristic is effective even without any external model.
>
> **(3) Practical usage guideline.**
> Practically, this leads to a simple trade-off:
> - for **high-precision evaluation or deployment**, one may use the calibrated BGS variant (with a one-time offline calibration per model);
> - for **lightweight or external-dependency-sensitive scenarios**, the **First50 heuristic** provides a fully self-contained alternative with almost the same behavior.

---

> ### Author Response · Authors · 2025-11-22
> **Response to Reviewer eqQN 3/5**
>
> > **Weakness 3: On possible failure modes of the Qwen2.5-VL-32B evaluator**
>
> **Response:** We appreciate the reviewer’s question about the reliability and potential failure modes of the VQA evaluator (Qwen2.5-VL-32B). We agree that understanding the behavior of the evaluation model is crucial.
>
> **(1) Quantitative error analysis**
>
> To systematically probe the evaluator, we manually annotated OmitI2V samples generated by frampack v1 and computed TP/FP/TN/FN statistics for each edit type:
>
> Table: Error statistics of the Qwen2.5-VL-32B evaluator on OmitI2V (FramePack V1). We report the FP rate, FN rate, and overall error for each edit type.
>
> | **Main category** | **FP rate** | **FN rate** | **Overall error** |
> |-------------------|------------:|------------:|-------------------:|
> | Addition          | 0.78%       | 1.94%       | 2.71%              |
> | Deletion          | 0.92%       | 3.15%       | 4.07%              |
> | Modification      | 0.63%       | 2.76%       | 3.38%              |
> | **All**           | **0.77%**   | **2.61%**   | **3.38%**          |
>
>
> These results indicate that Qwen2.5-VL-32B is generally reliable on this task, with low error rates across all three edit types.
>
> **(2) Observed systematic tendencies**
>
> When we inspect the mistakes, we do observe two mild, but interpretable, tendencies:
>
> - **False negatives on small or partially occluded objects (conservative behavior).**
>   In some *addition* and *deletion* clips, the evaluator answers *“no”* to object-presence questions even though the object is present but **small, partially hidden, or overshadowed by a larger foreground object**.
>   Example pattern:
>   > Question: “Is a cat visible in the video?”
>   > GT: Yes;
>   > Model: No, there is no cat visible in the video. The video shows a bear walking through a valley, not a cat.
>   In such cases, Qwen2.5-VL-32B seems to focus on the dominant object (e.g., a larger animal) and misses the smaller cat, leading to a conservative “no”.
>
> - **Occasional over-endorsement of the prompt effect (slight “positive” bias).**
>   In a few *modification* cases, the evaluator’s answer is semantically plausible but slightly **overstates the strength of the edit**.
> > Question: “Does the instructor fade out of view while still holding the beaker?”
>   > GT: No;
>   > Model: Yes. The instructor, who is wearing a white lab coat and blue gloves, is seen holding a beaker filled with a green liquid. As the video progresses, the instructor gradually fades out of view while still holding the beaker, indicating that they are no longer visible but the beaker remains in their hand. This suggests that the instructor's presence is being removed from the scene, possibly to focus attention on the beaker or to transition to another part of the demonstration.
>   In such cases, the model correctly detects some visual change, but **hallucinates a stronger effect** than truly rendered.
>
> **(3) Confidence in semantic adherence measurement**
>
> To complement this analysis:
>
> - We show in the paper (and supplementary) that Qwen2.5-VL-32B’s scores **correlate well with human MOS** in our user study, which supports its validity as a proxy for semantic adherence on OmitI2V.
> - The error modes above are mostly **borderline or subtle cases** (tiny objects, very mild appearance changes), whereas the majority of OmitI2V edits involve clear semantic operations (add / remove / modify an object), where the evaluator behaves consistently.
> - Recent works have leveraged MLLMs to assess prompt following and semantic adherence in related settings, including image editing [1] and subject-driven image generation [2].
>
> Based on the above analysis, we are confident that the VQA-based evaluation provides a reliable and nuanced measure of semantic adherence for OmitI2V.
>
> [1] Yang Ye, Xianyi He, Zongjian Li, et al. Imgedit: A unified image editing dataset and benchmark. arXiv preprint arXiv:2505.20275, 2025.
> [2] Chenyuan Wu, Pengfei Zheng, Ruiran Yan, et al. Omnigen2: Exploration to advanced multimodal generation. arXiv preprint arXiv:2506.18871, 2025.

---

> ### Author Response · Authors · 2025-11-22
> **Response to Reviewer eqQN 4/5**
>
> > **Question 1: On empirically validating that ASM reduces attention entropy**
>
> **Response:** We appreciate the reviewer’s request for empirical validation of our theoretical analysis. We fully agree that it is important to verify that the performance gains of AlignVid are indeed rooted in the predicted attention dynamics.
>
> **(1) Measuring attention entropy in the targeted blocks/steps.**
> In the revised manuscript, we explicitly measure attention entropy *during generation* in exactly the blocks and steps where ASM is applied:
>
> - We focus on **video queries** and consider their attention distribution over the **conditioning block** (text + image tokens) in the blocks selected by BGS and within the step range where SGS activates ASM.
> - For each such query, head, and layer, we compute the Shannon entropy of this conditioning attention **before** and **after** applying AlignVid, and report the ratio \(H_{\text{after}} / H_{\text{before}}\), averaged over queries and OmitI2V samples.
>
> This analysis is presented in the *Attention Analysis* subsection and visualized in Figure 3 of the revised paper.
>
> **(2) Empirical confirmation of entropy reduction and “softmax sharpening”.**
> Consistent with our temperature-based analysis in Section 4, we observe that after applying ASM:
>
> - The distribution of \(H_{\text{after}} / H_{\text{before}}\) is **systematically below 1** in the targeted blocks, indicating that attention within the conditioning block becomes lower-entropy and more concentrated.
> - At the same time, the **maximum attention scores** per query increase, confirming that the distribution becomes “sharper” rather than merely rescaled.
>
> In other words, AlignVid empirically realizes the predicted **softmax sharpening** effect: attention mass is pushed toward a small set of high-logit tokens, and away from many weak ones.
>
> **(3) Correlation with semantic fidelity.**
> We further examine how this entropy reduction relates to semantic fidelity:
>
> - At the **signal level**, after ASM the video queries allocate **stronger attention to text tokens**, and **relatively less attention to image regions**. This aligns with our “signal imbalance” view, where the image prior acts as a resistance signal and the text/temporal tokens carry the edit signal.
> - At the **metric level**, configurations that exhibit a **larger average entropy reduction** in the conditioning block also achieve **higher semantic fidelity** on OmitI2V. In particular, the AlignVid variant shows both **lower average conditioning-block entropy** and **higher edit-success rates** than the same backbone without ASM.
>
> In summary, we have (i) directly measured attention entropy in the blocks/steps where ASM is active, (ii) confirmed that AlignVid reduces this entropy in line with the theoretical temperature view, and (iii) shown that such entropy reduction is accompanied by a reallocation of attention toward prompt-relevant cues and by higher semantic fidelity scores. We have made this analysis explicit, with quantitative statistics and visualizations, in the revised manuscript.

---

> ### Author Response · Authors · 2025-11-22
> **Response to Reviewer eqQN 5/5**
>
> > **Question 2: Why does adaptive energy-based modulation underperform simple scalar scaling?**
>
> **Response:** We appreciate the reviewer’s question regarding why the more complex adaptive “energy-based modulation” performs worse than simple scalar scaling (Table 3). Our intuition is that, in this setting, the **simplicity and consistency of a fixed scalar multiplier** are actually more suitable than a dynamically adapting factor, because the underlying problem—semantic negligence caused by signal conflict—requires a **strong, stable bias** rather than a fragile, time-varying one.
>
> **(1) The need for a consistent, systemic bias**
>
> The core challenge in TI2V semantic negligence is a **systemic conflict**: across frames and denoising steps, the high-energy image condition (the *resistance signal*) tends to overpower the text condition (the *edit signal*).
>
> - **Fixed scalar scaling (\(\alpha > 1\)).**
>   By applying a single, well-chosen fixed scalar, AlignVid provides a **consistent and reliable gain** to the conditioning signal. This strong, unwavering bias helps the edit signal remain influential throughout the whole generation process, repeatedly counteracting the image “preserve” tendency. In other words, the problem calls for a blunt but robust tool to address a persistent, global conflict.
>
> - **Adaptive modulation.**
>   In contrast, the adaptive scheme attempts to compute a precise \(\alpha\) from the current logit distribution. In practice, this factor can be **too conservative or too volatile**: it may briefly apply a high gain, but can also drop back toward \(\alpha \approx 1\) even when the conflict is still strong, allowing the system to fall back to the image’s “preserve” command and causing the edit to fail.
>
> **(2) Instability and noise sensitivity of adaptive scaling**
>
> The inputs used to derive any adaptive scaling factor (e.g., maximum logit, variance, or global attention energy) are inherently **noisy** in diffusion, especially in the crucial early and mid denoising steps.
>
> - **Instability.**
>   The adaptive factor \(\alpha_{\text{adaptive}}\) becomes a function of noisy quantities (\(Q\), \(K\), and logits). This dependence injects **additional variance** into the attention distribution. In video generation, where temporal coherence and smoothness are critical, a constantly fluctuating \(\alpha\) can harm stability.
>
> - **Loss of predictability.**
>   Dynamic modulation risks applying **insufficient gain** (leading to semantic failure) or **excessive, uncalibrated gain** (leading to over-concentrated, near one-hot attention) at the wrong steps, which can destroy subtle cues required for coherent videos.
>
> In summary, while adaptive scaling is theoretically more elegant, our experiments show that the **simple, well-calibrated fixed scalar** is empirically superior for AlignVid:
>
> - it provides a **stable, high-gain bias** that consistently counteracts the systemic image–text conflict,
> - avoids the instability and noise sensitivity of the current energy-based design, and
> - is also **computationally cheaper**, since it only adds scalar multiplications on \(Q/K\), whereas the energy-based variant needs to aggregate attention statistics.

---

> > ### Comment · Reviewer_eqQN · 2025-11-26
> > **Reply to rebuttal**
> >
> > Thanks authors for the detailed response. It addressed my concerns. Therefore, I raised my rating.

---

> > > ### Author Response · Authors · 2025-11-26
> > > **Thank you again**
> > >
> > > Dear Reviewer eqQN,
> > >
> > > Thank you again for taking the time to review our paper and for your invaluable comments. Your feedback has been extremely helpful in improving the quality and clarity of our work, and we truly appreciate your updated evaluation.
> > >
> > > Best regards,
> > > Authors

---

### Official Review · Reviewer_o7F5 · 2025-11-07

**Soundness:** 2
**Presentation:** 3
**Contribution:** 2
**Rating:** 4
**Confidence:** 4

**Summary:**

This paper proposes to address the issue of poor instruction following in Text-Guided Image-to-Video (TI2V) generation (semantic negligence as noted by authors). This occurs when models frequently fail to incorporate complex semantic changes by the text prompt (e.g., object addition or deletion).

The core motivation comes from a study showing that applying Gaussian blur to the input image unexpectedly improves instruction following. Through visualizing the attention maps, the authors was able to see clear foreground and background separation. The authors link this effect to achieving a lower-entropy attention distribution (i.e., clearer foreground-background separation) within the model.

To exploit this finding without sacrificing quality, the authors propose AlignVid, a training-free framework. AlignVid deploys two components (i) Attention Scaling Modulation (ASM) which scales the Q/K to directly modulate the attention and (ii) Guidance Scheduling (GS) which determines where to apply ASM.

Finally, the paper introduces OmitI2V, a new human-annotated benchmark focused specifically on quantitative evaluation of instruction following (semantic adherence) across addition, deletion, and modification scenarios. Results reportedly demonstrate the effectiveness of this approach.

**Strengths:**

* Theoretically Motivated Approach: The authors provide an excellent theoretical foundation, analyzing attention maps to observe clear foreground–background separation. They effectively link this empirical finding to an energy perspective, correctly identifying the mechanism as corresponding to a desirable lower-entropy attention distribution.

* Excellent Writing and Clarity: The paper is well-written, professional, and easy to follow.

* Extensive Validation Across Baselines: The effectiveness of the method is demonstrated through comprehensive studies across multiple state-of-the-art baselines (specifically FramePack, FramePack F1, and Wan2.1). The consistent and significant improvements shown in the OmitI2V benchmark validate the proposed approach.

**Weaknesses:**

* Lack of Established Evaluation Benchmarks: The proposed AlignVid method is exclusively validated using the proposed OmitI2V benchmark. Including evaluation results on established public benchmarks would significantly strengthen the paper's claims and demonstrate generalizability. For example, the ViCLIP metric [1], which assesses video-text semantic alignment, would be particularly useful to include.

* Lack of Generalization Experiments: The proposed AlignVid approach sounds like a general-purpose method that could also be applicable to standard Text-to-Image (T2I) and Text-to-Video (T2V) generation settings, as it operates on the attention mechanism. It would be valuable to see evaluation results demonstrating the effectiveness of AlignVid within these broader contexts.

[1] Internvid: A large-scale video-text dataset for multimodal understanding and generation, ICLR 2023.

**Questions:**

1. Comparison with Classifier-Free Guidance (CFG): Could the authors provide a comparison between the proposed AlignVid approach and Classifier-Free Guidance (CFG)? Since CFG is the widely adopted technique to enhance controllability in generative models, a thorough analysis of how AlignVid performs (either alone or when combined with CFG) is essential for justifying its contribution.

2. Generalization to Broader Tasks: Given that the AlignVid approach operates by modifying the attention mechanism, it appears to be a generalizable technique. Can the authors discuss and provide experimental results demonstrating the effectiveness of AlignVid when applied to standard T2I and T2V generation tasks?

---

> ### Author Response · Authors · 2025-11-22
> **Response to Reviewer o7F5 1/3**
>
> > **Weakness 1: Lack of Established Evaluation Benchmarks**
>
> **Response:** Thank you for highlighting the importance of evaluating with established metrics. We have added the ViCLIP metric [1] to the manuscript. Specifically, we report ViCLIP scores for all baselines and for AlignVid in the updated **Tables 2–5** of the manuscript. The results show that AlignVid consistently improves ViCLIP score, indicating that our attention modulation not only alleviates semantic negligence on OmitI2V but also enhances global text–video alignment under a public, standardized metric. We have updated **Section 6** to describe the ViCLIP evaluation setup.
>
> **Here we show Table 2 in the manuscript: Effectiveness of our method.** Values in parentheses indicate relative improvement over the corresponding baseline. Our method consistently boosts semantic alignment and motion dynamics with only marginal changes in aesthetic quality.
>
> | Method              | SemAlign Mod ↑ | SemAlign Add ↑ | SemAlign Del ↑ | ViCLIP Mod ↑ | ViCLIP Add ↑ | ViCLIP Del ↑ | Dynamic Degree ↑ | Aesthetic Quality ↑ |
> |---------------------|----------------|----------------|----------------|--------------|--------------|--------------|-------------------|---------------------|
> | FramePack           | 64.99          | 68.55          | 58.14          | 20.83        | 21.08        | 20.43        | 20.05             | 63.94               |
> | FramePack + Ours    | 68.22 (+3.23)  | 73.13 (+4.58)  | 60.21 (+2.07)  | 21.25 (+0.42)| 22.08 (+0.83)| 20.86 (+0.43)| 28.53 (+8.48)     | 63.57 (−0.37)       |
> | FramePack F1        | 64.45          | 67.79          | 58.50          | 21.06        | 19.91        | 20.61        | 24.42             | 63.10               |
> | FramePack F1 + Ours | 71.27 (+6.82)  | 71.60 (+3.81)  | 61.06 (+2.56)  | 21.78 (+0.72)| 21.04 (+1.13)| 20.99 (+0.38)| 33.16 (+8.74)     | 62.10 (−1.00)       |
> | Wan2.1              | 72.35          | 71.75          | 63.13          | 20.93        | 20.59        | 20.82        | 46.02             | 63.12               |
> | Wan2.1 + Ours       | 77.20 (+4.85)  | 79.54 (+7.79)  | 69.47 (+6.34)  | 22.19 (+1.26)| 23.30 (+2.71)| 21.29 (+0.47)| 47.04 (+1.02)     | 61.63 (−1.49)       |
>
> [1] Internvid: A large-scale video-text dataset for multimodal understanding and generation, ICLR 2023.

---

> ### Author Response · Authors · 2025-11-22
> **Response to Reviewer o7F5 2/3**
>
> > **Weakness 2: Lack of Generalization Experiments (T2I / T2V)**
>
> **Response:** We appreciate the reviewer’s suggestion to validate AlignVid beyond the TI2V setting. To assess its generality, we have added quantitative experiments  and visualization comparison on **T2I, T2V, and image editing benchmarks**:
>
> - **T2I compositional generation (GenEval).**
>   The new GenEval results (see Table GenEval) are reported *without* using the prompt rewriter. AlignVid improves most compositional metrics, including **Single object**, **Two object**, **Colors**, **Position**, and **Color attribution**, and **aesthetic score**. This shows that the same attention modulation helps standard text-to-image generation, not only TI2V generation.
>
> **Table: Quantitative results on GenEval (prompt rewriter not used).**
>
> | Method        | Single object ↑ | Two object ↑ | Counting ↑ | Colors ↑ | Position ↑ | Color attribution ↑ | Aesthetic Score ↑ |
> |--------------|-----------------|--------------|------------|----------|------------|----------------------|-------------------|
> | OmniGen2     | 0.99            | 0.94         | 0.67       | 0.85     | 0.55       | 0.62                 | 5.517             |
> | + AlignVid   | 1.00 (+0.01)    | 0.97 (+0.03) | 0.52 (-0.15) | 0.89 (+0.04) | 0.60 (+0.05) | 0.70 (+0.08)        | 5.568 (+0.05)     |
>
> - **General T2V quality (VBench).**
>   The VBench results (Table VBench) show that AlignVid yields gains overally, such as **Spatial Relationship** (+11.51), **Scene** (+6.18), and **Multiple Objects** (+13.11), as well as modest improvements in **Subject Consistency** and **Motion Smoothness**. Some metrics (e.g., **Dynamic Degree**, **Aesthetic Quality**) slightly decrease, but overall the pattern indicates that AlignVid improves global text–video alignment and structural correctness in the T2V evaluation.
>
> **Table: Quantitative results on VBench. AlignVid yields gains in the T2V task.**
>
> | Model      | Subject Consistency ↑ | Temporal Style ↑ | Temporal Flickering ↑ | Spatial Relationship ↑ | Scene ↑ | Overall Consistency ↑ | Object Class ↑ | Multiple Objects ↑ | Motion Smoothness ↑ | Imaging Quality ↑ | Dynamic Degree ↑ | Color ↑ | Background Consistency ↑ | Appearance Style ↑ | Aesthetic Quality ↑ |
> |-----------|------------------------|------------------|------------------------|------------------------|---------|------------------------|----------------|---------------------|---------------------|-------------------|------------------|--------|--------------------------|--------------------|---------------------|
> | Wan2.1-T2V-1.3B  | 94.24                  | 22.67            | 99.32                  | 72.74                  | 19.62   | 23.59                  | 79.03          | 53.35               | 97.77               | 69.70             | 70.83            | 88.08  | 98.09                    | 19.58              | 64.60               |
> | + AlignVid| 94.51 (+0.27)          | 23.46 (+0.79)    | 98.66 (-0.66)          | 84.25 (+11.51)         | 25.80 (+6.18) | 24.47 (+0.88)    | 79.91 (+0.88)  | 66.46 (+13.11)      | 98.05 (+0.28)      | 68.53 (-1.17)     | 68.06 (-2.77)    | 91.80 (+3.72) | 98.20 (+0.11)            | 20.16 (+0.58)      | 62.69 (-1.91)       |
>
>
> - **Image Editing task (ImgEdit).**
>   On the ImgEdit benchmark (Table ImgEdit), when applied to OmniGen2 in the image editing setting, AlignVid improves **Add**, **Remove**, **Style**, and **Aesthetic Score**. This confirms that the same attention-scaling strategy generalizes to editing tasks.
>
> **Table: Quantitative results on ImgEdit. AlignVid yields gains in the image editing task.**
>
> | Model      | Add ↑   | Adjust ↑ | Extract ↑ | Replace ↑ | Remove ↑ | Background ↑ | Style ↑ | Compose ↑ | Action ↑ | Aesthetic Score ↑ |
> |-----------|---------|----------|-----------|-----------|----------|--------------|---------|-----------|----------|--------------------|
> | OmniGen2  | 2.52    | 3.27     | 2.08      | 3.12      | 2.83     | 3.65         | 4.57    | 2.89      | 4.59     | 5.606              |
> | + AlignVid| 3.53 (+1.01) | 3.12 (-0.15) | 2.04 (-0.04) | 3.18 (+0.06) | 3.33 (+0.50) | 3.65         | 4.75 (+0.18) | 2.43 (-0.46) | 4.50 (-0.09) | 5.624 (+0.02)     |
>
> Together, these results demonstrate that AlignVid is not limited to our OmitI2V TI2V benchmark: the same training-free attention modulation provides consistent semantic gains on **standard T2V (VBench), compositional T2V generation (GenEval), and Image editing (ImgEdit)**.

---

> ### Author Response · Authors · 2025-11-22
> **Response to Reviewer o7F5 3/3**
>
> > **Question 1: Comparison with Classifier-Free Guidance (CFG)**
>
> **Response:** Thank you for raising this important point. We agree that a clear comparison with classifier-free guidance (CFG) is essential, since CFG is the de facto mechanism for improving controllability in diffusion models. We compare CFG with AlignVid as follows:
>
> **(1) Conceptual relationship between CFG and AlignVid.**
> CFG operates at the *score level*: it mixes conditional and unconditional predictions to globally steer the denoising direction. In contrast, AlignVid acts *inside* the transformer, by rescaling attention logits for specific token groups (text/image/video). In other words, CFG changes the predicted noise, while AlignVid redistributes attention mass across modalities. This makes the two mechanisms complementary rather than mutually exclusive.
>
> **(2) Quantitative comparison on Wan2.1 with different CFG scales.**
> To analyze this relationship empirically, we evaluated Wan2.1 under both weak and strong CFG, and then applied AlignVid on top:
>
> **Table: Comparison with CFG on Wan2.1.**
>
> | Method              | SemAlign Mod ↑ | SemAlign Add ↑ | SemAlign Del ↑ | ViCLIP Mod ↑ | ViCLIP Add ↑ | ViCLIP Del ↑ | Dynamic Degree ↑ | Aesthetic Quality ↑ |
> |---------------------|----------------|----------------|----------------|--------------|--------------|--------------|-------------------|---------------------|
> | CFG = 1 (no cfg)    | 63.55          | 63.66          | 61.06          | 19.20        | 17.09        | 19.67        | 41.65             | 61.19               |
> | CFG = 1 + AlignVid  | 65.88          | 72.52          | 60.21          | 19.51        | 18.47        | 19.89        | 42.48             | 62.11               |
> | CFG = 5 (Official)  | 72.35          | 71.75          | 63.13          | 20.83        | 21.08        | 20.43        | 46.02             | 63.12               |
> | CFG = 5 + AlignVid  | 77.20          | 79.54          | 69.47          | 22.19        | 23.30        | 21.29        | 47.04             | 61.63               |
>
> AlignVid and CFG are complementary: applying AlignVid on top of CFG consistently boosts semantic alignment across all edit types for both weak guidance (CFG=1) and strong guidance (CFG=5), while maintaining comparable visual quality.
> - **CFG = 1 (weak guidance).**
>   Compared to `CFG = 1 (no cfg)`, adding AlignVid (`CFG = 1 + AlignVid`) improves semantic alignment scores from
>   - Modification: 63.55 → 65.88
>   - Addition: 63.66 → 72.52
>   while keeping Deletion comparable (61.06 → 60.21) and slightly improving visual quality (Dynamic Degree: 41.65 → 42.48, Aesthetic: 61.19 → 62.11). ViCLIP scores also increase across all edit types (e.g., Addition: 17.09 → 18.47).
>
> - **CFG = 5 (official setting).**
>   Under the recommended guidance scale, AlignVid again brings clear gains:
>   - Semantic alignment: 72.35 / 71.75 / 63.13 → 77.20 / 79.54 / 69.47 (Modification / Addition / Deletion)
>   - ViCLIP: 20.83 / 21.08 / 20.43 → 22.19 / 23.30 / 21.29
>   Dynamic Degree slightly improves (46.02 → 47.04), while Aesthetic Quality remains at a comparable level (63.12 → 61.63).
>
> These results show that AlignVid consistently improves semantic alignment and ViCLIP scores **both with weak guidance (CFG = 1) and strong guidance (CFG = 5)**.
>
> **(3) Computational and implementation differences.**
> CFG also requires training the model with an explicit unconditional branch and doubles the number of forward passes at inference time (conditional + unconditional) for guided sampling. In contrast, AlignVid is entirely training-free and only introduces lightweight scalar multiplications on attention queries/keys, without additional network evaluations or architecture changes. This makes AlignVid particularly attractive as a low-overhead plug-in for existing DiT-based generators, including models whose training recipe (and CFG setup) is fixed or inaccessible.
>
> Overall, CFG and AlignVid address controllability at different levels: CFG adjusts the global denoising trajectory, whereas AlignVid explicitly rebalances cross-modal attention (text, image, video). The experiments demonstrate that AlignVid does not replace CFG but *complements* it—adding AlignVid on top of standard CFG yields additional semantic gains, while remaining training-free and computationally lightweight. This supports the contribution of AlignVid as a general, plug-and-play attention modulation mechanism rather than an alternative to CFG.

---

### Author Response · Authors · 2025-12-01
**Author Final Remarks to AC**

Dear Area Chair,

We sincerely appreciate your time and effort for the community. To assist in your assessment and save valuable time, we provide a detailed summary of our rebuttal process below.

We appreciate that reviewers **o7F5**, **eqQN**, and **MrsV** converge on several key strengths of our work. (1) Reviewers **eqQN** and **MrsV** agree that we clearly formalize a critical and prevalent weakness in TI2V—semantic negligence—and that the OmitI2V benchmark is a valuable contribution specifically targeting addition, deletion, and modification, which is missing in existing benchmarks.  (2)  Reviewers **o7F5** and **eqQN** both emphasize the solid theoretical motivation of our method. (3) Reviewers **o7F5** and **eqQN** highlight our extensive validation across strong baselines.  (4) Reviewer **MrsV** further notes that AlignVid is simple, practical, and requires no retraining to be applied to existing TI2V models. (5) Reviewer **o7F5** also commends the excellent writing and clarity. This convergent positive feedback from all three reviewers reinforces the significance and potential impact of our contributions.

The main concerns are: 1) **Evaluation and generalization** – reviewer **o7F5** asks for results on other metircs (e.g., ViCLIP) and for evidence that AlignVid generalizes beyond OmitI2V to standard T2I/T2V settings, as well as comparisons to CFG; 2) **Dependence on external evaluators/pipelines** – reviewer **eqQN** is concerned about relying on MLLM as the primary semantic metric and on a PCA+SAM2 calibration pipeline for block selection, and requests compare BGS with simpler block-selection strategies; 3) **Theory–practice connection and trade-offs** – reviewer **MrsV** questions how our entropy-based analysis is tied to conditioned TI2V, asks why a new benchmark is needed instead of extending VBench, and highlights the trade-off between improved semantic adherence and slightly reduced aesthetic quality, requesting more quantitative results (e.g., entropy measurements).

**1. Rebuttal and paper revision**

**1) More evaluation and generalization experiments (reviewer **o7F5**)**
   - We add **ViCLIP** evaluations for all TI2V models and show that AlignVid consistently improves video–text semantic alignment.
   - We add **T2I, T2V, and image-editing** experiments (GenEval, VBench, ImgEdit) and visualization results, showing that AlignVid generalizes beyond OmitI2V.
   - We compare AlignVid against Classifier-Free Guidance (CFG) and show that AlignVid is **complementary**: adding AlignVid on top of both weak and strong CFG further improves semantic fidelity.


**2) Explain on external evaluators/pipelines （reviewer **eqQN**）**
   - We provide an analysis of the MLLM evaluator across edit types, with overall error around 3.38%, and discuss its main (interpretable) failure modes.
   - We add **attention analyses** (figures and statistics) showing that AlignVid indeed reduces conditioning-block entropy, reallocates attention mass toward text, and that this correlates with improved semantic scores.
   - We add ablations for **Block-level Guidance Scheduling (BGS)** (foreground-sensitive blocks vs. first-half vs. last-half blocks).
   - We analyze why the **energy-based** adaptive modulation underperforms simple scalar scaling and position it as an exploratory variant, recommending **scalar scaling** as the default due to its stability and effectiveness.

**3) Explain the theory–practice connection and trade-offs （reviewer **MrsV**）**
   - We rewrite the theory section to explicitly model *text/image/video* token groups and to link entropy reduction to the observed attention imbalance (over-reliance on the image prior and under-weighting text).
   - We clarify that VBench-I2V mainly measures how well a model preserves the input subject/background. OmitI2V is a similarly sized but specialized benchmark that focuses on high-conflict edit scenarios to evaluate semantic negligence rather than simple fidelity directly.
   - We clarify that in *video* tasks (TI2V / T2V), AlignVid trades a small amount of aesthetic score for significantly better semantic alignment and motion. However, on **T2I and image editing** tasks, AlignVid can improve both **semantic fidelity** *and* **aesthetics**, suggesting that the trade-off is mainly tied to motion-related artifacts in video rather than an inherent loss of quality.
   - We add attention statistics over multiple OmitI2V examples, comparing attention scores and entropies **before and after applying a Gaussian blur**. The results show that blur consistently strengthens text-related attention and reduces conditioning-block entropy, quantitatively supporting our pilot observation.

**2. Reviewers' reply**:

Reviewer **eqQN** stated that we had addressed their concerns and raised the score.

Finally, we sincerely thank all reviewers for their thoughtful comments and are deeply grateful to the Area Chair for your time and effort. Thank you!

Best regards,

Authors

---

### Note · Program_Chairs · 2026-01-17
**Submission Desk Rejected by Program Chairs**

The following references in this submission do not refer to real documents and/or have major errors in bibliographic information:

 Shengyu Liu, Yifan Zhu, Yang Zhou, et al. Video-p2p: Video editing with cross-attention control. In Proceedings of the IEEE/CVF Conference on Computer Vision and Pattern Recognition (CVPR), pp. 12164-12173, 2024b.